# Associations of gamma-glutamyl transferase with cardio-metabolic diseases in people living with HIV infection in South Africa

Kim A. Nguyen[1]*, Nasheeta Peer[1,2], Andre P. Kengne[1,2]

1 Non-Communicable Diseases Research Unit, South African Medical Research Council, Cape Town and Durban, South Africa, 2 Department of Medicine, University of Cape Town, Cape Town, South Africa

* Kim.Nguyen@mrc.ac.za

**Data Availability Statement:** Relevant data can be found at http://medat.samrc.ac.za/index.php/catalog/43.

## Abstract

### Background

Gamma-glutamyl transferase (GGT) has recently been reported as a biomarker for cardio-vascular (CVD) risk in general populations. We investigated the associations of GGT with cardio-metabolic diseases and CVD risk in South Africans living with HIV.

### Methods

In this cross-sectional study, HIV-infected adults were randomly recruited across 17 HIV clinics in the Western Cape Province. Homeostatic model assessment for insulin resistance (HOMA-IR), hypertension, diabetes, metabolic syndrome by Joint Interim Statement criteria (JIS-MS), a ≥5% and ≥10% predicted risk for a CVD event within 10 years by the Framingham risk score (10-years-CVD risk) were computed. Associations between GGT and cardio-metabolic trait were explored using linear and binomial logistic regressions adjusted for age, gender, lifestyle behaviours and HIV-related characteristics.

### Results

Among 709 participants (561 women, mean age 38.6 years), log-GGT was positively associated with waist circumference (β=2.75; p<0.001), diastolic blood pressure (β=1.65; p=0.006), total cholesterol (β=0.21; p<0.001), low-density lipoprotein-cholesterol (β=0.16; p<0.001), high-density lipoprotein-cholesterol and log-triglycerides (both β=0.12; p<0.001), fasting plasma glucose (β=0.19; p=0.031), 2-hour-post-glucose-load plasma glucose (β=0.26; p=0.007), HOMA-IR (β=0.13; p=0.001), log-high-sensitivity C-reactive-protein (β=0.3; p<0.001) in linear regression analyses; with hypertension [OR=1.41 (95% CI, 1.13-1.75); p=0.001], JIS-MS [OR=1.33 (1.05-1.68); p=0.016], ≥5% 10-year-CVD risk [OR=1.55 (1.24-1.9400); p<0.001] and ≥10% 10-year-CVD risk [OR=1.56 (1.08-2.23); p=0.016] but not with diabetes [OR=1.24 (0.88-1.71), p=0.205] in logistic regression analyses.

**Funding:** The work reported herein was made possible through funding by the South African Medical Research Council through its Division of Research Capacity Development under the SAMRC Intramural Postdoctoral Programme from funding received from the South African National Treasury. The content hereof is the sole responsibility of the authors and do not necessarily represent the official views of the SAMRC or the funders.

**Competing interests:** The authors have declared that no competing interests exist.

## Conclusions

In this study, GGT levels were associated with cardio-metabolic variables independent of HIV specific attributes. If confirmed in longitudinal studies, GGT evaluation maybe included in CVD risk monitoring strategies in people living with HIV.

## Introduction

Cardiovascular diseases (CVD), which claim 38 million lives annually, are the leading cause of mortality worldwide, including in adults infected with human immunodeficiency virus (HIV) [1]. The rise of CVD in HIV-infected populations has followed the introduction and widespread uptake of antiretroviral treatment (ART) and the associated improved survival and longevity in these individuals [2, 3]. Consequently, the natural progression of HIV infection has shifted from the fatal acquired immunodeficiency syndrome (AIDS) to a manageable chronic condition, with non-AIDS-defining illnesses increasingly responsible for morbidity and mortality in HIV-infected people [4–6]. Notably, CVD and metabolic diseases are among the most frequent non-infectious co-morbidities in HIV-infected individuals; these likely arise from the combined effects of HIV infection, extended exposure to ART, and traditional CVD risk factors associated with ageing [7]. These fuel recommendations for early detection and treatment of cardio-metabolic diseases in HIV-infected people at high risk for CVD [8].

Considering that the development of CVDs in the HIV-infected are attributable to both traditional risk factors and HIV-related variables, findings in general populations may not be generalisable to the HIV-infected. Therefore, there is a need to determine whether cardio-metabolic diagnostic tests/ biomarkers recommended in general populations are applicable to HIV-infected populations. However, there is a paucity of evidence confirming such findings [9, 10]. For example, the association between gamma-glutamyl transferase (GGT) and CVD risk, while confirmed in general populations, has been little studied in the HIV-infected [11, 12]. GGT, a liver enzyme which is present in the serum and on the surface of most cell membranes, has long been recommended as a biomarker of hepatobiliary disease and excessive alcohol consumption [13]. GGT has also been related to cardio-metabolic diseases such as obesity, hypertension and type 2 diabetes mellitus (hereafter referred to as diabetes), and elevated GGT levels predict the development of metabolic syndrome, CVD events and mortality [14, 15]. Furthermore, the relationship of GGT concentrations with cardio-metabolic diseases were observed even within GGT normal range [16, 17], and without concomitantly increased levels of other liver enzymes [18]. Therefore, the present study aims to examine the associations of GGT with CVD risk factors, insulin resistance and Framingham risk scores (FRS).

## Materials and methods

### Study design and population

This cross-sectional study recruited HIV-infected adults from 17 public healthcare facilities in Cape Town and the surrounding rural municipalities. Simple random sampling technique was used to select the participants that has been described in detail previously [19, 20]. Participants were ≥18-year-old HIV-positive men and women who were not pregnant, breastfeeding, bedridden, undergoing treatment for cancer, nor on corticosteroid treatment.

The study was approved by the South African Medical Research Council Ethics Committee (Official Letter no. EC021-11/2013), and by the Health Research Committee of the Western

Cape Department of Health (Document no. RP 005/2014). All participants provided written informed consents for their participation in the study.

## Data collection

A trained research team including clinicians, nurses and fieldworkers conducted the data collection using electronic case report forms with built-in checks for quality control [21]. The data collected were captured on personal digital assistants (PDAs) onto a web-based respondent driven sampling research management system. Simultaneously, participants' data were linked and tracked via a unique barcode using BRYANT Research systems software.

Socio-demographic data, including lifestyle behaviour and medical history, were obtained from a structured interviewer-administered questionnaire adapted from the WHO STEPwise approach to Surveillance (STEPS) tool. HIV-related information such as duration of diagnosed HIV infection, CD4 counts and ART were obtained from the participants' clinical records.

**Measurements.** Anthropometry was collected using standardised techniques; heights and weights were measured with participants wearing light clothing and without shoes. Waist circumference (WC) was measured at the level of umbilicus. Blood pressure (BP) was taken with a digital BP monitor (Omron, M6 Comfort, Netherland) on the right arm while the participant was seated and had rested for at least 5 minutes. BP was taken thrice at three minutes intervals, and the average of $2^{nd}$ and $3^{rd}$ readings used in the analysis.

All participants without previously diagnosed diabetes underwent a standard 2-hour 75-gram oral glucose tolerance test (OGTT) after an overnight fast. Plasma glucose levels were assessed at fasting (FPG) and at 2-hour post-OGTT (2h-PG). Blood samples were drawn and processed for laboratory analyses. The concentrations of glucose and lipid were measured with an autoanalyser, Beckman Coulter AU 500 spectrophotometer. Serum GGT, serum cholesterol and triglycerides were analysed using enzymatic colorimetric method and high-sensitivity C-reactive protein (hs-CRP) was read using immunoturbidimetric system. Glycated hemoglobin (HbA1c) was measured using high-performance liquid chromatography (VARIANT II TURBO. EDTA tubes) following the National Glycohaemoglobin Standardisation Programme (NGSP) certified according to Roche Diagnostics [22]. Insulin concentrations were determined with Chemiluminescence Immunoassay. Homeostatic model assessment for insulin resistance (HOMA-IR) was calculated as insulin (mIU/L) times glucose (mmol/L) and divides by 22.5 (Insulin X glucose / 22.5) [23].

**Definitions.** The following categories were defined: 1) Current smokers as those who currently smoked any tobacco products such as cigarettes, cigars or pipes daily or occasionally; 2) Current drinkers: consumed at least one alcoholic drink at any occasion during the previous 30 days; 3) Hypertension: systolic BP (SBP) $\geq$140 mmHg and/or diastolic BP (DBP) $\geq$90 mmHg and/or a history of using hypertensive medication; 4) Diabetes: FPG $\geq$7.0 mmol/L and/or 2h-PG $\geq$11.1 mmol/L and/or history of using diabetes medication; 5) Metabolic syndrome based on the Joint Interim Statement criteria (JIS-MS) [24]: increased WC: men $\geq$94 cm, women $\geq$80 cm; high triglycerides: $\geq$1.7 mmol/L; low high-density lipoprotein cholesterol (HDL-C): men <1.03 mmol/L, women <1.3 mmol/L; raised BP: $\geq$130/85 mmHg or on hypertensive medication; hyperglycemia: FPG $\geq$5.6 mmol/L or on glucose control agents; 6) A $\geq$5% and $\geq$10% predicted risk for CVD within 10 years by FRS (10-years-FRS-CVD risk) [25]. Accounting for age, gender, cholesterol levels, SBP, smoking and diabetes status, the FRS identifies individuals at higher risk for CVD and has been recommended for use in primary care setting [26].

## Statistical analysis

Data were analysed using R statistical software version 3.6.0 (2019-04-26). Continuous variables were summarized as means (standard deviation, SD) or medians ($25^{th}$ to $75^{th}$

percentiles), and categorical variables as count (percentages). Baseline characteristics were compared across GGT groups, which were defined by quartiles of GGT levels (GGT-Q1 being the lowest quarter and GGT-Q4 the highest quarter). These were done using chi-square tests, fisher-exact tests, t-tests or Kruskal-Wallis tests for non-parametric data or Analysis of Variance tests (ANOVA) where appropriate. The Spearman correlation test was used to explore the correlations between GGT levels and baseline variables while Levene and Cochrane-Armitage trend tests were used to investigate linear trend across GGT quarters.

Associations between GGT and cardio-metabolic trait were explored using linear and binomial logistic regressions adjusted for 1) age and gender; 2) smoking and alcohol consumption; 3) duration of HIV-diagnosis; 4) age, gender, smoking, alcohol consumption and HIV-diagnosed duration pooled models; C-statistics were computed. Skewed variables were log-transformed to approximate normal distribution before performing regression analyses. A secondary analysis was done in a sub-set of participants with data available on CD4 counts, to explore the likely effects of these attributes on the relationship of GGT with cardio-metabolic risk. For a z-value of 1.96 (corresponding to a 95% confidence interval), and a sample size of 709 participants, our study had a margin of error of 0.07% to detect a prevalence of hypertension of 1%. This indicated our study was well-power as the accepted margin is of 5%.

## Results

The study recruited 831 participants; however, some participants (n=77) did not return for biochemical assessments and some (n=45) blood samples were insufficient for analysis. The present analyses therefore included 709 participants (561 women and 148 men) whose characteristics across GGT quarters are shown in Table 1. Mean levels of SBP, DBP, total cholesterol, triglycerides, HDL-C, low-density lipoprotein cholesterol (LDL-C), HbA1c, FPG, 2h-PG and hs-CRP generally differed across GGT quarters (all $p \leq 0.035$). However, there were linear increases in the trend for mean total cholesterol, triglycerides, HDL-C, HbA1c, FPG, 2h-PG, and median fasting insulin and HOMA-IR across increasing GGT quarters (all $p \leq 0.048$ for linear trend).

Age, gender, smoking, alcohol intake, hypertension, diabetes, JIS-MS and $\geq 5\%$ 10-year-CVD risk differed across GGT quarters (all $p \leq 0.032$). Hypertension, diabetes, JIS-MS FRS-10-year-CVD risk $\geq 5\%$ and $\geq 10\%$, ART use, current smoking and current drinking increased linearly across increasing GGT quarters (all $p \leq 0.044$). The distribution of women across GGT quarters decreased linearly (p<0.001). GGT was positively correlated with age, WC, SBP, DBP, LDL-C and hs-CRP (all $p \leq 0.012$ for Spearman correlations).

### Associations of GGT and cardio-metabolic risk factors

In age and sex adjusted linear regression models, GGT was associated with WC, DBP, total cholesterol, LDL-C, HDL-C, triglycerides, FPG, 2h-PG, fasting insulin, HOMA-IR and hs-CRP (Table 2). In the fully adjusted model, these associations still remained significant (all $p \leq 0.031$). The association of GGT with fasting insulin was also significant in the fully adjusted model (p=0.004). GGT was not associated with BMI or HbA1c ($p \geq 0.063$).

In the logistic regressions (Table 3), GGT was significantly associated with hypertension [OR 1.44 (95%CI 1.16-1.78)] and MS by JIS criteria [OR 1.25 (95%CI 1.00-1.55)] in the age and sex adjusted models, and in the fully adjusted models. However, the association with diabetes was not significant in the age and sex adjusted model nor in the fully adjusted model.

Tables 4 and 5 present the same linear and logistic regression models in the subset of participants in whom CD4 count data were available, adjusted additionally for this variable. The patterns of associations were mostly similar to the models described above without CD4 counts.

**Table 1. Characteristics of participants categorised by Gamma-Glutamyl Transferase (GGT) quarters.**

| Characteristics | Overall | GGT quarters Q1 | Q2 | Q3 | Q4 | P-trend P-value | Spearman correlation rho | P-value |
|---|---|---|---|---|---|---|---|---|
| N | 709 | 173 | 169 | 188 | 179 | | | |
| Median GGT (P25-P75) | 39 (26-67) | 20 (16-23) | 31 (28-35) | 49 (43-56) | 112 (80-207) | <0.001 | NA | NA NA |
| Mean age, years (SD) | 38.6 (9.0) | 37 (8.5) | 37.6 (9.2) | 40.1 (9.4) | 39.6 (8.7) | 0.001 | 0.485 | 0.13 0.001 |
| Women, n (%) | 561 (79.1) | 148 (85.5) | 142 (84) | 152 (80.8) | 119 (66.5) | <0.001 | <0.001 | -0.17 <0.001 |
| Median CD4 count (P25-P75) | 395 (240-600) | 400 (252-598) | 397 (256-619) | 428 (238-666) | 344 (194-499) | 0.128 | 0.239 | -0.07 0.168 |
| Median HIV-duration, years (P25-P75) | 5 (2-9) | 5.0 (2.0-8.3) | 5.0 (3.0-9.0) | 5.0 (2.0-9.0) | 5.0 (2.0-8.0) | 0.643 | 0.955 | -0.03 0.385 |
| ART users, n (%) | 617/661 (93.3) | 147/164 (89.6) | 146/153 (95.4) | 164/180 (91.1) | 160/164 (97.6) | 0.659 | 0.026 | 0.04 0.229 |
| Mean body mass index (SD) | 27.6 (6.9) | 27.7 (6.8) | 27.6 (6.8) | 27.9 (7.0) | 27.3 (7.1) | 0.821 | 0.816 | -0.02 0.626 |
| Mean waist circumference (SD) | 89.0 (14.4) | 87.2 (13.9) | 88.0 (14.9) | 90.5 (14.6) | 90.4 (14.1) | 0.064 | 0.709 | 0.09 0.010 |
| Current smokers, n (%) | 179 (25.2) | 37 (21.4) | 39 (23.1) | 45 (23.9) | 58 (32.4) | 0.078 | 0.021 | -0.09 0.019 |
| Current drinkers, n (%) | 187 (26.4) | 21 (12.1) | 37 (21.9) | 47 (25) | 82 (45.8) | <0.001 | <0.001 | 0.26 <0.001 |
| Mean systolic blood pressure (SBP) (SD) | 120.8 (19.8) | 116.9 (19.6) | 119.1 (18.4) | 122.1 (19.1) | 124.7 (24.0) | 0.001 | 0.496 | 0.17 <0.001 |
| Mean diastolic blood pressure (DBP) (SD) | 83.5 (12.6) | 80.9 (12.1) | 82.1 (11.6) | 84.0 (12.6) | 86.5 (13.2) | <0.001 | 0.321 | 0.17 <0.001 |
| Mean total cholesterol (SD) | 4.4 (1.0) | 4.2 (1.0) | 4.2 (0.9) | 4.5 (1.1) | 4.7 (1.1) | <0.001 | 0.033 | 0.16 <0.001 |
| Median triglycerides (P25-P75) | 0.99 (0.75-1.34) | 0.92 (0.68-1.19) | 0.89 (0.72-1.24) | 1.02 (0.76-1.38) | 1.15 (0.84-1.46) | <0.001 | 0.016 | 0.20 <0.001 |
| Mean LDL-C (SD) | 2.57 (0.89) | 2.45 (0.84) | 2.49 (0.83) | 2.61 (0.96) | 2.71 (0.92) | 0.026 | 0.183 | 0.09 0.012 |
| Mean HDL-C (SD) | 1.33 (0.41) | 1.25 (0.33) | 1.22 (0.29) | 1.36 (0.42) | 1.5 (0.49) | <0.001 | <0.001 | 0.22 <0.001 |
| Mean Glycated hemoglobin (HbA1c) (SD) | 5.6 (0.8) | 5.5 (0.8) | 5.4 (0.5) | 5.7 (1.1) | 5.6 (0.8) | 0.029 | 0.006 | 0.07 0.043 |
| Mean fasting plasma glucose (FPG) (SD) | 5.5 (1.7) | 5.3 (1.9) | 5.1 (1.1) | 5.6 (2.1) | 5.6 (1.7) | 0.035 | 0.004 | 0.18 <0.001 |
| Mean 2h-glucose (SD) | 5.6 (1.9) | 5.3 (1.3) | 5.4 (1.8) | 5.8 (2.3) | 5.8 (1.9) | 0.017 | 0.027 | 0.11 0.005 |
| Median fasting insulin (P25-P75) | 6.0 (3.8-9.0) | 5.6 (4.0-7.9) | 5.9 (4.0-8.7) | 6.7 (4.3-11.1) | 5.8 (3.3-9.8) | 0.067 | 0.048 | 0.04 0.306 |
| Median HOMA-IR (P25-P75) | 1.4 (0.8-2.2) | 1.2 (0.8-1.8) | 1.3 (0.8-1.9) | 1.5 (0.9-2.6) | 1.4 (0.8-2.5) | 0.054 | 0.005 | 0.06 0.118 |
| Median hs-CRP (P25-P75) | 5.6 (2.4-11.8) | 3.6 (1.5-8.2) | 5.6 (2.4-12.8) | 6.4 (2.8-14.9) | 7.0 (3.1-17.4) | 0.000 | 0.366 | 0.18 <0.001 |
| Median creatinine (P25-P75) | 58 (51-66) | 58 (51-65) | 57 (49-65) | 57 (51-65) | 59 (52-69) | 0.130 | 0.423 | 0.06 0.125 |
| Median ALT (P25-P75) | 23 (17-34) | 18 (14-22) | 20 (16-26) | 25 (19-31) | 35 (24-50) | <0.001 | <0.001 | 0.50 <0.001 |
| Median AST (P25-P75) | 29 (24-38) | 25 (21-30) | 26 (22-31) | 30 (26-38) | 38 (29-48) | <0.001 | <0.001 | 0.44 <0.001 |
| [1]Hypertension, n (%) | 258 (36.4) | 47 (27.2) | 48 (28.4) | 79 (42) | 84 (46.9) | <0.001 | <0.001 | 0.17 <0.001 |
| [2]Diabetes, n (%) | 62 (8.7) | 10 (5.8) | 9 (5.3) | 23 (12.2) | 20 (11.2) | 0.032 | 0.015 | 0.09 0.015 |
| [3]JIS-MS, n (%) | 200 (28.2) | 40 (23.1) | 46 (27.2) | 56 (29.8) | 58 (32.4) | 0.253 | 0.044 | 0.08 0.045 |
| Framingham risk score (10-year predicted CVD risk ≥5%), n (%) | 172 (24.3) | 28 (16.2) | 33 (19.6) | 49 (26.1) | 62 (34.6) | 0.005 | 0.001 | 0.13 0.001 |
| Framingham risk score (10-year predicted CVD risk ≥10%), n (%) | 43 (6.1) | 6 (3.5) | 8 (4.7) | 15 (7.9) | 14 (7.8) | 0.191 | 0.042 | 0.08 0.043 |

GGT-Q1, GGT<26IU/L; GGT-Q2, 26IU/L≤GGT<39IU/L; GGT-Q3, 39IU/L≤GGT<67IU/L; GGT-Q4, GGT≥67IU/L; LDL-C, low-density lipoprotein cholesterol; HDL-C, high-density lipoprotein cholesterol; HOMA-IR, homeostatic model assessment for insulin resistance; hs-CRP, high-sensitivity C-reactive protein; ALT, alanine aminotransferase; AST, aspartate aminotransferase

[1]Hypertension, SBP≥140mmHg and/or DBP≥90mmHg and/or history of hypertension

[2]Diabetes, FPG≥7.0mmol/L and/or 2h-PG≥11.1mmol/L and/or history of diabetes

[3]JIS-MS, metabolic syndrome based on Joint Interim Statement (2009) criteria.

**Table 2. Linear regressions for the associations of Gamma-Glutamyl Transferase (GGT) with cardio-metabolic risk markers, N=709.**

| Predictors / Outcomes | Log GGT | Log GGT + Age & Gender | Log GGT + Smoking & Alcohol drinking | Log GGT + HIV-duration | Log GGT + Age, gender, smoking, alcohol drinking & HIV- duration |
|---|---|---|---|---|---|
| BMI | -0.27 (0.415) | 0.33 (0.272) | 0.24 (0.438) | -0.17 (0.601) | 0.56 (0.063) |
| WC | 1.40 (0.039) | 2.13 (0.001) | 2.45 (<0.001) | 1.68 (0.012) | 2.75 (<0.001) |
| SBP | 3.27 (<0.001) | 1.70 (0.052) | 2.90 (0.002) | 3.47 (<0.001) | 1.37 (0.130) |
| DBP | 2.29 (<0.001) | 1.87 (0.001) | 2.06 (<0.001) | 2.45 (<0.001) | 1.65 (0.006) |
| TC | 0.25 (<0.001) | 0.26 (<0.001) | 0.27 (<0.001) | 0.26 (<0.001) | 0.21 (<0.004) |
| LDL-C | 0.14 (0.001) | 0.14 (0.001) | 0.17 (<0.001) | 0.14 (0.001) | 0.16 (<0.001) |
| HDL-C | 0.13 (<0.001) | 0.14 (<0.001) | 0.11 (<0.001) | 0.14 (<0.001) | 0.12 (<0.001) |
| Log TG | 0.12 (<0.001) | 0.11 (<0.001) | 0.13 (<0.001) | 0.13 (<0.001) | 0.12 (<0.001) |
| HbA1c | 0.04 (0.361) | 0.02 (0.596) | 0.07 (0.104) | 0.03 (0.398) | 0.04 (0.339) |
| FPG | 0.20 (0.015) | 0.17 (0.045) | 0.23 (0.006) | 0.20 (0.015) | 0.19 (0.031) |
| 2h-OGTT | 0.28 (0.002) | 0.26 (0.006) | 0.35 (<0.001) | 0.28 (0.003) | 0.26 (0.007) |
| Log fasting insulin | 0.01 (0.729) | 0.06 (0.055) | 0.06 (0.083) | 0.01 (0.659) | 0.10 (0.004) |
| Log HOMA-IR | 0.05 (0.206) | 0.09 (0.011) | 0.10 (0.008) | 0.05 (0.177) | 0.13 (0.001) |
| Log hs-CRP | 0.29 (<0.001) | 0.29 (<0.001) | 0.31 (<0.001) | 0.29 (<0.001) | 0.30 (<0.001) |

Data are β coefficient and p-value (in log transformation); BMI, body mass index; WC, waist circumference; SBP, systolic blood pressure; DBP, diastolic blood pressure; TC, total cholesterol; LDL-C, low density lipoprotein cholesterol; HDL-C, high density lipoprotein cholesterol; TG, triglycerides; HbA1c, glycated hemoglobin; FPG, fasting plasma glucose; 2h-OGTT, 2 hour-oral glucose tolerant test; HOMA-IR, homeostatic model assessment for insulin resistance; hs-CRP, high sensitivity c-reactive protein.

## Discussion

To our knowledge, there is an on-going cohort study by Strijdom et al. aiming to determine whether HIV-infection and ART are associated with cardiovascular risk variables and changes in vascular endothelial structure and function in adults living in South Africa [11, 12]. A recently published paper described higher GGT levels in HIV-infected versus HIV-uninfected Africans but did not examine the associations of GGT with cardio-metabolic diseases [12]. The main findings in our study were that serum GGT was positively associated with WC, BP, hypertension, dyslipidaemia, dysglycaemia, insulin resistance, hs-CRP and metabolic syndrome by JIS criteria after adjusting for age, sex, smoking, alcohol consumption and HIV-related factors. However, no association was found with BMI, HbA1c or diabetes. We also demonstrated the correlation of GGT with a ≥5% and ≥10% 10-years-CVD risk by the FRS.

These findings demonstrate that the associations of serum GGT with cardio-metabolic variables in an HIV-infected population accord with the literature on general populations [14, 27, 28]. In the latter, serum GGT has been reported to be a biomarker for visceral adiposity, hepatic steatosis, insulin resistance, metabolic syndrome and risk of CVD morbidity and mortality. Nevertheless, the evidence from the literature on the associations of serum GGT with cardio-metabolic diseases in general populations cannot be generalised to HIV-infected populations. The associations in the HIV-infected may differ because of the different pathways involved in the development of cardio-metabolic diseases in the HIV-infected, i.e., traditional risk factors together with HIV-specific factors, which could differentially influence the relation of GGT and cardio-metabolic outcomes [29–31].

### GGT and abdominal obesity

Although several studies suggest that GGT correlates with BMI defined obesity [32, 33], this was not observed in our study; however, GGT levels were positively associated with WC. The

**Table 3. Logistic regressions for the associations of gamma-glutamyl transferase (GGT) with cardio-metabolic diseases and Framingham cardiovascular risk, N=709.**

| Variables | OR (95%CI) | P-value | C-statistic |
|---|---|---|---|
| **Hypertension[1]** | | | |
| n=258 | | | |
| Log GGT | 1.10 (1.04-1.14) | 0.001 | 0.598 |
| + Age, gender | 1.44 (1.16-1.78) | 0.001 | 0.714 |
| + Smoking, alcohol drinking | 1.45 (1.20-1.78) | 0.001 | 0.596 |
| + HIV-duration | 1.10 (1.05-1.15) | 0.001 | 0.610 |
| + Age, gender, smoking, alcohol drinking and HIV-duration | 1.41 (1.13-1.75) | 0.001 | 0.728 |
| **Diabetes[2]** | | | |
| n=62 | | | |
| Log GGT | 1.03 (0.99-1.04) | 0.202 | 0.572 |
| + Age, gender | 1.30 (0.93-1.78) | 0.302 | 0.672 |
| + Smoking, alcohol drinking | 1.33 (0.97-1.80) | 0.073 | 0.565 |
| + HIV-duration | 1.03 (1.00-1.06) | 0.037 | 0.578 |
| + Age, gender, smoking, alcohol drinking and HIV-duration | 1.24 (0.88-1.71) | 0.205 | 0.674 |
| **Metabolic Syndrome[3]** | | | |
| n=200 | | | |
| Log GGT | 1.04 (0.99-1.08) | 0.086 | 0.548 |
| + Age, gender | 1.25 (1.00-1.55) | 0.047 | 0.679 |
| + Smoking, alcohol drinking | 1.31 (1.05-1.61) | 0.014 | 0.593 |
| + HIV-duration | 1.04 (1.00-1.09) | 0.041 | 0.608 |
| + Age, gender, smoking, alcohol drinking and HIV-duration | 1.33 (1.05-1.68) | 0.016 | 0.692 |
| **FRS-10- year CVD risk ≥5%[4]** | | | |
| n=172 | | | |
| Log GGT | 1.55 (1.26-1.91) | <0.001 | 0.605 |
| + Alcohol drinking | 1.53 (1.23-1.91) | <0.001 | 0.604 |
| + HIV-duration | 1.57 (1.27-1.94) | <0.001 | 0.610 |
| + Alcohol drinking and HIV-duration | 1.55 (1.24-1.94) | <0.001 | 0.610 |
| **FRS-10- year CVD risk ≥10%** | | | |
| n=43 | | | |
| Log GGT | 1.03 (1.01-1.06) | 0.004 | 0.605 |
| + Alcohol drinking | 1.56 (1.08-2.23) | 0.015 | 0.607 |
| + HIV-duration | 1.03 (1.01-1.06) | 0.004 | 0.603 |
| + Alcohol drinking and HIV-duration | 1.56 (1.08-2.23) | 0.016 | 0.605 |

Odds ratios (OR) and 95% confidence interval (CI) from logistic regression models adjusted for age, gender, current smoking, current alcohol drinking and HIV-diagnosed duration

[1]Hypertension, SBP≥140mmHg and/or DBP≥90mmHg and/or history of hypertension

[2]Diabetes, FPG≥7mmol/L and/or 2h-PG≥11.1mmol/L and/or history of diabetes; MS (JIS), metabolic syndrome based on Joint Interim Statement (2009) criteria; 10-year predicted cardiovascular disease risk by Framingham Risk Score.

association of GGT with WC and not BMI is in keeping with a recent cross-sectional study among Chinese adults [34]. This suggests that visceral adipose tissue, as measured by WC, better correlates with GGT than subcutaneous adipose tissue; the latter identified by BMI and representing general adiposity. Notably, WC as an indicator of visceral or central obesity, better identifies CVD risk [24]. Furthermore, raised WC is closely related to fatty liver disease which

**Table 4. Linear regressions for the associations of gamma-glutamyl transferase (GGT) with cardio-metabolic risk markers, based on a subset of participants whose CD4 count data were available, N=357.**

| Predictors<br><br>Outcomes | Log GGT | Log GGT + Age & Gender | Log GGT + Smoking & Alcohol drinking | Log GGT + CD4 count & HIV-duration | Log GGT + Age, gender, smoking, alcohol drinking, CD4 count & HIV-duration |
|---|---|---|---|---|---|
| BMI | -0.39 (0.406) | 0.27 (0.551) | 0.15 (0.740) | -0.10 (0.824) | 0.53 (0.236) |
| WC | 0.84 (0.364) | 1.52 (0.099) | 1.81 (0.051) | 1.43 (0.116) | 2.01 (0.026) |
| SBP | 2.80 (0.029) | 1.69 (0.174) | 2.93 (0.030) | 3.11 (0.016) | 1.70 (0.190) |
| DBP | 2.07 (0.015) | 1.98 (0.021) | 2.15 (0.016) | 2.32 (0.006) | 1.98 (0.022) |
| TC | 0.23 (0.001) | 0.23 (0.001) | 0.25 (0.001) | 0.22 (0.001) | 0.23 (0.001) |
| LDL-C | 0.11 (0.070) | 0.11 (0.064) | 0.14 (0.028) | 0.11 (0.075) | 0.13 (0.046) |
| HDL-C | 0.12 ($<$0.001) | 0.13 ($<$0.001) | 0.09 ($<$0.001) | 0.12 ($<$0.001) | 0.09 ($<$0.001) |
| Log TG | 0.14 ($<$0.001) | 0.13 ($<$0.001) | 0.15 ($<$0.001) | 0.14 ($<$0.001) | 0.13 ($<$0.001) |
| HbA1c | -0.00 (0.984) | -0.01 (0.808) | 0.04 (0.575) | 0.01 (0.839) | 0.01 (0.905) |
| FPG | 0.19 (0.086) | 0.17 (0.130) | 0.24 (0.033) | 0.22 (0.039) | 0.29 (0.072) |
| 2h-OGTT | 0.36 (0.010) | 0.34 (0.018) | 0.48 (0.001) | 0.41 (0.003) | 0.44 (0.002) |
| Log Fasting insulin | 0.02 (0.599) | 0.07 (0.133) | 0.06 (0.193) | 0.04 (0.398) | 0.10 (0.036) |
| Log HOMA | 0.06 (0.232) | 0.10 (0.047) | 0.11 (0.044) | 0.08 (0.110) | 0.14 (0.009) |
| Log hs-CRP | 0.30 (0.001) | 0.31 (0.001) | 0.35 ($<$0.001) | 0.31 ($<$0.001) | 0.36 ($<$0.001) |

Data are β coefficient and P-value (in log transformation); BMI, body mass index; WC, waist circumference; SBP, systolic blood pressure; DBP, diastolic blood pressure; TC, total cholesterol; LDL-c, low density lipoprotein cholesterol; HDL-C, high density lipoprotein cholesterol; TG, triglycerides; FPG, fasting plasma glucose; 2h-OGTT, 2 hour-oral glucose tolerant test; HOMA, homeostasis model assessment; hs-CRP, high sensitivity c-reactive protein.

is a risk factor for future CVD independent of BMI [35]. Therefore, the association of GGT with WC in this study may likely suggest greater CVD risk in participants with raised GGT.

## GGT and hypertension

The positive associations of GGT with both systolic and diastolic BP and hypertension in our study are supported by cross-sectional and longitudinal studies that describe the relationship of GGT with hypertension in general populations [36, 37]. Shankar and Li reported that higher serum GGT levels were positively associated with prehypertension in adults without a history of hypertension and CVD in the National Health and Nutrition Examination Survey (NHANES) 1999-2002 [38]. The positive relationship between GGT and hypertension was also reported in a population-based study among Chinese adults; interestingly, this relationship was stronger in those with increased WC [36].

**Table 5. Logistic regressions for the associations of gamma-glutamyl transferase (GGT) with cardio-metabolic diseases and Framingham cardiovascular risk based on a subset of participants whose CD4 count data were available, N=375.**

| Variables | OR (95%CI) | P-value | C-statistic |
|---|---|---|---|
| **Hypertension[1]** | | | |
| n=127 | | | |
| Log GGT | 1.09 (1.02-1.16) | 0.007 | 0.593 |
| + Age, gender | 1.44 (1.06-1.98) | 0.021 | 0.717 |
| + Smoking, alcohol drinking | 1.49 (1.11-2.01) | 0.008 | 0.598 |
| + HIV-duration, CD4 count | 1.10 (1.03-1.17) | 0.003 | 0.608 |
| + Age, gender, smoking, alcohol drinking, HIV-duration and CD4 count | 1.47 (1.06-2.04) | 0.021 | 0.733 |
| **Diabetes[2]** | | | |
| n=26 | | | |
| Log GGT | 1.02 (0.99-1.05) | 0.273 | 0.639 |
| + Age, gender | 1.29 (0.72-2.24) | 0.360 | 0.707 |
| + Smoking, alcohol drinking | 2.33 (0.77-7.78) | 0.141 | 0.675 |
| + HIV-duration, CD4 count | 1.04 (1.01-1.08) | 0.022 | 0.654 |
| + Age, gender, smoking, alcohol drinking, HIV-duration and CD4 count | 1.34 (0.71-2.41) | 0.347 | 0.737 |
| **Metabolic Syndrome[3]** | | | |
| n=106 | | | |
| Log GGT | 1.05 (0.98-1.11) | 0.147 | 0.553 |
| + Age, gender | 1.31 (0.96-1.78) | 0.082 | 0.641 |
| + Smoking, alcohol drinking | 1.40 (1.04-1.91) | 0.029 | 0.609 |
| + HIV-duration, CD4 count | 1.06 (0.99-1.13) | 0.052 | 0.606 |
| + Age, gender, smoking, alcohol drinking, HIV-duration and CD4 count | 1.48 (1.07-2.06) | 0.018 | 0.663 |
| **FRS-10- year CVD risk ≥5%[4]** | | | |
| n=70 | | | |
| Log GGT | 1.84 (1.33-2.55) | <0.001 | 0.654 |
| + Alcohol drinking | 1.84 (1.31-2.59) | <0.001 | 0.654 |
| + HIV-duration, CD4 count | 1.86 (1.34-2.61) | <0.001 | 0.656 |
| + Alcohol drinking, HIV-duration and CD4 count | 1.86 (1.32-2.65) | <0.001 | 0.656 |
| **FRS-10- year CVD risk ≥10%** | | | |
| n=16 | | | |
| Log GGT | 1.03 (1.00-1.06) | 0.027 | 0.653 |
| + Alcohol drinking | 1.86 (1.01-3.35) | 0.040 | 0.653 |
| + HIV-duration, CD4 count | 1.03 (1.00-1.06) | 0.037 | 0.657 |
| + Alcohol drinking, HIV-duration and CD4 count | 1.76 (0.96-3.16) | 0.058 | 0.657 |

Odds ratios (OR) and 95% confidence interval (CI) from logistic regression models adjusted for age, gender, smoking, alcohol drinking, HIV-diagnosed duration and CD4 count

[1]Hypertension, SBP≥140mmHg and/or DBP≥90mmHg and/or history of hypertension

[2]Diabetes, FPG≥7mmol/L and/or 2h-PG≥11.1mmol/L and/or history of diabetes

[3]MS (JIS), metabolic syndrome based on Joint Interim Statement (2009) criteria

[4]10-year predicted cardiovascular disease risk by Framingham Risk Score.

## GGT and insulin resistance and diabetes

GGT has been associated with insulin resistance and diabetes in cross-sectional and cohort studies in general populations [39]. While we observed a significant association between GGT and HOMA-IR, there was no relation between GGT and diabetes. However, the latter needs to

be viewed with caution due to a lack of precision attributed to the small number of diabetes cases (n=62) in our sample.

### GGT and metabolic syndrome

Similar to our results, a South African study conducted in a mixed-ancestry community found significant associations between GGT and insulin resistance and the metabolic syndrome; the latter is one of few studies which has assessed these relationships in an African population [39]. In the Framingham Heart Study, an increase in serum GGT predicted the incidence of metabolic syndrome, and CVD events and deaths among 3541 participants [27].

### GGT and Framingham risk score

To our knowledge, no other study has specifically investigated the association of GGT with the Framingham CVD risk equation (FRS). FRS criteria was developed to predict CVD risk levels over 10-year period using individual CVD risk markers including age, sex, smoking, blood lipids, BP and diabetes status. Our finding of a positive association between GGT and $\geq 5\%$ and $\geq 10\%$ 10-years FRS-CVD risk after adjusting for alcohol consumption and HIV-related characteristics suggests the need for further exploration of this association. Future studies may examine serum GGT as a predictor of CVD risk level in both HIV-infected and general populations.

### Pathophysiological pathways of the cardio-metabolic associations with GGT

In general populations, GGT is associated with fatty liver disease and insulin resistance and is a predictor of hypertension, diabetes and metabolic syndrome [32, 40, 41]; however, the pathophysiological mechanisms linking GGT with cardio-metabolic risk are not fully explained. Nonetheless, the presence of GGT together with oxidized LDL-C and foam cells within atherosclerotic plaques suggest the direct participation of GGT in atherosclerotic formation [42]. The postulated pathways may include the role of GGT in the following: 1) low-grade inflammation i.e. it is proinflammatory; 2) oxidative stress and 3) free radical production [43]. The significant associations of GGT with LDL-C and hs-CRP, which are proinflammatory atherogenic markers, support the assumption that GGT is involved in inflammatory and oxidative stress pathways.

### Strengths and limitations

The strength of this study is that it addresses a gap in the literature by appraising the associations of GGT with CVD risk marker in an HIV-infected population, which to our knowledge, has not been previously examined. However, considering the cross-sectional study design, causality could not be established for the associations of GGT with CVD risk markers. The small number of men in this study, a limitation of many South African studies, suggests that our results are likely driven by women who form majority of this sample, and thus our results may not be generalisable to both genders. Furthermore, the adjustment for HIV specific attributes was only partial; we could not control for the possible effects of antiretroviral drugs and co-infections with hepatitis B or hepatitis C. An increasing risk of hepatotoxicity has been reported in HIV patients on ART and with the latter co-infections [44]. The data on hepatitis B or C co-infections and antiretroviral drug class were not collected and precluded these analyses; the association of serum GGT level with the D:A:D equation which is specific for CVD risk estimation in people with HIV could not be appraised. Lastly, the small number of cases

for diabetes precluded reliable analysis due to the low statistical power; the FRS criteria which was developed in white population might not be applicable to black populations.

## Conclusions

This study found GGT levels to be associated with cardio-metabolic variables in HIV-infected adults, independent of HIV specific attributes. More research in HIV-infected populations, especially studies with larger sample sizes and more men are required. If our results are confirmed in other studies, it would support the potential use of GGT as a cardiovascular biomarker in people with HIV independent of its utility in liver disease. It must be emphasised that HIV-infected patients regularly frequent healthcare services and, given that a large proportion are at risk for CVD, routine CVD risk screening and lifestyle modifications should be encouraged in this population.

## Author Contributions

**Conceptualization:** Kim A. Nguyen, Nasheeta Peer, Andre P. Kengne.

**Formal analysis:** Kim A. Nguyen, Andre P. Kengne.

**Funding acquisition:** Kim A. Nguyen, Nasheeta Peer.

**Methodology:** Kim A. Nguyen, Andre P. Kengne.

**Supervision:** Nasheeta Peer.

**Writing – original draft:** Kim A. Nguyen.

**Writing – review & editing:** Nasheeta Peer, Andre P. Kengne.

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
