## [Decision Letter · Decision Letter 0]

29 Oct 2020

PONE-D-20-26437

Associations of gamma-glutamyl transferase with cardio-metabolic diseases in people living with HIV infection in South Africa

PLOS ONE

Dear Dr. Nguyen,

Thank you for submitting your manuscript to PLOS ONE. After careful consideration, we feel that it has merit but does not fully meet PLOS ONE’s publication criteria as it currently stands. Therefore, we invite you to submit a revised version of the manuscript that addresses the points raised during the review process.

From my own reading of the manuscript, I agree with the reviewer's comments. Please consider each carefully and incorporate as you see fit. I look forward to receiving your revised manuscript.

We look forward to receiving your revised manuscript.

Kind regards,

Ethan Morgan

Academic Editor

PLOS ONE

Journal Requirements:

2. Please provide further details on sample size and power calculations.

3. In statistical methods, please clarify whether you corrected for multiple comparisons.

4.In your Data Availability statement, you have not specified where the minimal data set underlying the results described in your manuscript can be found. PLOS defines a study's minimal data set as the underlying data used to reach the conclusions drawn in the manuscript and any additional data required to replicate the reported study findings in their entirety. All PLOS journals require that the minimal data set be made fully available. For more information about our data policy, please see http://journals.plos.org/plosone/s/data-availability.

Reviewers' comments:

Reviewer's Responses to Questions

**Comments to the Author**

1. Is the manuscript technically sound, and do the data support the conclusions?

Reviewer #1: Yes

Reviewer #2: Partly

2. Has the statistical analysis been performed appropriately and rigorously? 

Reviewer #1: I Don't Know

Reviewer #2: Yes

3. Have the authors made all data underlying the findings in their manuscript fully available?

Reviewer #1: Yes

Reviewer #2: Yes

4. Is the manuscript presented in an intelligible fashion and written in standard English?

Reviewer #1: Yes

Reviewer #2: Yes

5. Review Comments to the Author

Reviewer #1: Thank you for this detailed analysis. I notice the statistical effort in compiling the results. I am a clinician. I cant comment on the statistical methodology of the study. My comments are more from a clinician's perspective that I hope would make sense as follows.

Main comments:

1. The authors highlight that the association between GGT and metabolic syndrome/ CVD risk has already been established in general population. What would be the reason to expect such an association would not apply to PLWH?

2. I am sorry to say that I am not sure what would be the clinical use of the information of this meticulous analysis. Are we proposing that we should be screening patients for CVD/ DM/ etc when the GGT is raised? Or, should we dismiss further management of raised GGT when any of the identified conditions are present?

3. In practice, most of the patients with raised GGT have elevated ALT and AST too (indeed Table 1 also agrees with this statement). We interpret liver profile and not just one of the liver enzyme to define clinical abnormality. How do we propose to use the study information for management of those patients?

4. I am not sure the quarters of GGT used for the analysis are clinically relevant. The study's GGT quarters Q1, Q2 and Q3 are all within refence range and normal. Should we not carry out the analysis between all within reference range compared with those with elevated range according to the MACS classification of abnormal liver enzymes?

5. I notice the authors have acknowledged that they did not investigate the roles of viral hepatitis and cART medications. I think these are quite important omissions particularly considering the background prevalence of the infections in South Africa.

6. Unless I am mistaken, the analysis included even one unit of alcohol per time of study definition in the alcohol consumption group. This is strictly correct however not clinically helpful. Essentially what we need is to identify the threshold of consumption beyond which GGT would rise.

7. I notice that the majority of the study patients were women? Any reason for this? Would this have caused a bias?

8. As stated earlier, I am not a statistician. I am interested to understand how nearly every factor investigated in this study had significant association with raised GGT? Some of the findings don't make immediate sense to me; e.g. why would CD4 count have such a relationship with raised GGT?

Minor

1. Use of Framingham score in a population of non-White patients mostly living in rural areas may create a majour issue with the validity of the calculated CVD risk.

2. The results sections are full of mathematical evidence with little narrative on their clinical implications.

Reviewer #2: The manuscript entitled “Associations of gamma-glutamyl transferase with cardio-metabolic diseases in people living with HIV infection in South Africa” and authored by Nguyen et al., set out to investigate the associations of GGT with cardio-metabolic diseases and CVD risk in a group of South Africans living with HIV. To achieve this, the authors conducted a cross-sectional study which included randomly recruited HIV-infected adults. Homeostatic model assessment for insulin resistance, and associations between GGT and cardio-metabolic trait were determined. The authors observed associations of GGT with a number of cardiovascular disease risk factors, but not with diabetes, and concluded that GGT levels were associated with cardio-metabolic variables in the HIV-infected subjects but independent of HIV specific attributes, with the suggestion that GGT evaluation maybe included in CVD risk monitoring strategies in people living with HIV if the results is confirmed in a longitudinal study.

.

Comments:

1. The authors mentioned at the beginning of the Discussion that the study was the first ever to investigate the association of GGT with cardiovascular risk factors in HIV infected Africans (Lines 228-229 and 291-293). This is not correct; the EndoAfrica study in the Western Cape Province of South Africa has done a similar study already.

(1. https://bmcinfectdis.biomedcentral.com/articles/10.1186/s12879-020-05173-6)

(2. https://bmcinfectdis.biomedcentral.com/articles/10.1186/s12879-016-2158-y)

I will suggest that the authors take a look at the studies to see how to re-frame their study/story.

2. The authors also observed that the association of GGT with almost all the parameters that were investigated was similar to what has been shown in various populations; both disease conditions and control groups. This suggests that the study is not adding anything new to subject.

3. Referring to HIV infected patients/subjects as “a sample” is not correct and should be corrected throughout the manuscript (E.g. Lines 18 and 36-37 in the Abstract section).

4. Lines 93-94 in the Method section. The sentence does not indicate the number of times the BP was taken, which is important in this case.

5. Lines 99-102. The sentence is not clear and needs to be revised.

6. Lines 102-104. The sentence needs to be referenced.

7. Line 106. I will suggest that you either say "as previously described" or you would have to put the mathematical equation.

8. Lines 139-140. I will suggest that the first sentence in the Results section is moved to the Methods section.

9. There are minor typographical errors that needs to be corrected. E.g. Line 105. The word “Chemiluminescence” has an extra “e”.

6. PLOS authors have the option to publish the peer review history of their article (what does this mean?). If published, this will include your full peer review and any attached files.

Reviewer #1: **Yes: **Kaveh Manavi

Reviewer #2: No

---

## [Author Response · Author response to Decision Letter 0]

21 Dec 2020

Please see the rebuttal letter attached in this submission.

---

## [Decision Letter · Decision Letter 1]

31 Dec 2020

PONE-D-20-26437R1

Associations of gamma-glutamyl transferase with cardio-metabolic diseases in people living with HIV infection in South Africa

PLOS ONE

Dear Dr. Nguyen,

Thank you for submitting your manuscript to PLOS ONE. After careful consideration, we feel that it has merit but does not fully meet PLOS ONE’s publication criteria as it currently stands. Therefore, we invite you to submit a revised version of the manuscript that addresses the points raised during the review process.

The reviewer's suggest a few additional minor changes. Please attend to these, I look forward to receiving your revision in due time.

We look forward to receiving your revised manuscript.

Kind regards,

Ethan Morgan

Academic Editor

PLOS ONE

Reviewers' comments:

Reviewer's Responses to Questions

**Comments to the Author**

1. If the authors have adequately addressed your comments raised in a previous round of review and you feel that this manuscript is now acceptable for publication, you may indicate that here to bypass the “Comments to the Author” section, enter your conflict of interest statement in the “Confidential to Editor” section, and submit your "Accept" recommendation.

Reviewer #2: (No Response)

2. Is the manuscript technically sound, and do the data support the conclusions?

Reviewer #2: Partly

3. Has the statistical analysis been performed appropriately and rigorously? 

Reviewer #2: Yes

4. Have the authors made all data underlying the findings in their manuscript fully available?

Reviewer #2: Yes

5. Is the manuscript presented in an intelligible fashion and written in standard English?

Reviewer #2: Yes

6. Review Comments to the Author

Reviewer #2: The authors have addressed most of the comments that were raised, however, the major concern has not been addressed.

Even though the authors have incorporated the suggested EndoAfrica project references, they still have it in the Introduction section that “the association between gamma-glutamyl transferase (GGT) and CVD risk, while confirmed in general populations, has not been investigated in the HIV-infected“. I think this needs to be corrected. Secondly, I do not agree to the revisions made to the first sentence of the Discussion section which says that “this study is among the first to investigate the associations of serum GGT with CVD risk factors in HIV-infected Africans”. The study is not the first and the authors would just have to accept that fact.

7. PLOS authors have the option to publish the peer review history of their article (what does this mean?). If published, this will include your full peer review and any attached files.

Reviewer #2: No

---

## [Author Response · Author response to Decision Letter 1]

7 Jan 2021

Please see the rebuttal letter attached in this submission.

---

## [Editor Report · Decision Letter 2]

14 Jan 2021

Associations of gamma-glutamyl transferase with cardio-metabolic diseases in people living with HIV infection in South Africa

PONE-D-20-26437R2

Dear Dr. Nguyen,

We’re pleased to inform you that your manuscript has been judged scientifically suitable for publication and will be formally accepted for publication once it meets all outstanding technical requirements.

Kind regards,

Ethan Morgan

Academic Editor

PLOS ONE
---

## [Editor Report · Acceptance letter]

25 Jan 2021

PONE-D-20-26437R2 

Associations of gamma-glutamyl transferase with cardio-metabolic diseases in people living with HIV infection in South Africa 

Dear Dr. Nguyen:

I'm pleased to inform you that your manuscript has been deemed suitable for publication in PLOS ONE. Congratulations! Your manuscript is now with our production department. 

Kind regards, 

on behalf of

Dr. Ethan Morgan 

Academic Editor

PLOS ONE